# Generalized Category Discovery under the Long-Tailed Distribution

## Abstract

Generalized Category Discovery (GCD) is a challenging problem, which aims at discovering novel categories in unlabelled data by transferring the knowledge from the labelled data. Existing methods often assume a uniform distribution of categories, which is not representative of real-world data that typically exhibits a long-tailed distribution. In this paper, we address the problem of GCD under a long-tailed distribution. Our approach introduces a novel framework that tackles the challenges of biased classifier learning and imprecise class number estimation. We propose adaptive sample selection based on confidence and density, balancing the model's training distribution and mitigating bias. Additionally, we present a density-peak-based method for accurate class number estimation in long-tailed settings. Experimental results demonstrate the effectiveness of our approach in discovering novel categories and outperforming state-of-the-art methods.

## 1 Introduction

Over the past years, computer vision has shown significant advancements in tasks such as image recognition (He et al., 2016). Despite these progressions, artificial systems still face challenges in recognizing and categorizing visual information accurately in dynamic and complex environments. The visual information present in the real world is far more diverse and intricate than the benchmark datasets. To tackle this issue, researchers have directed their attention towards learning techniques that require minimal human intervention, such as the semi-supervised learning approach (Oliver et al., 2018). However, one limitation of most semi-supervised methods is the common assumption that the unlabelled dataset contains a set of categories with a small labelled dataset. This assumption is unrealistic as it is not possible to label all categories in the real world at once, not to mention the categories in the unlabelled dataset may grow over time. Therefore, Category Discovery (CD) emerges as a research problem, which gains increasing attention. Initially, the problem is studied as Novel Category Discovery (NCD) (Han et al., 2019), aiming to discover novel categories in the unlabelled data, assuming disjoint class spaces between labelled and unlabelled data. Later, the assumption is relaxed such that the unlabelled data also contains instances from the seen (labelled) classes, resulting in a more generalized setting called Generalized Category Discovery (GCD).

Although significant progress has been made in the field of GCD, existing methods (*e.g.*, (Wen et al., 2022; Vaze et al., 2022a; Zhao et al., 2023)) often assume a uniform distribution of categories, which does not accurately reflect the long-tailed distribution commonly observed in real-world data. This distribution is characterized by a few categories containing a significant number of examples (head classes), while the majority of categories have only a few instances (tail classes). In this paper, we consider the GCD under the more realistic long-tailed distribution (see Fig. 1). The primary challenge in this context lies in the potential bias towards the head classes during category discovery, making it difficult to identify and accurately recognize the tail classes.

To tackle this problem, we propose a novel framework that makes two key contributions. First, we introduce a method for adaptively selecting confident samples from the unlabelled dataset based on prediction confidence and density. These selected samples are used to form training mini-batches, effectively balancing the distribution of the model's training and reducing bias. We also aim to align the prediction distribution of

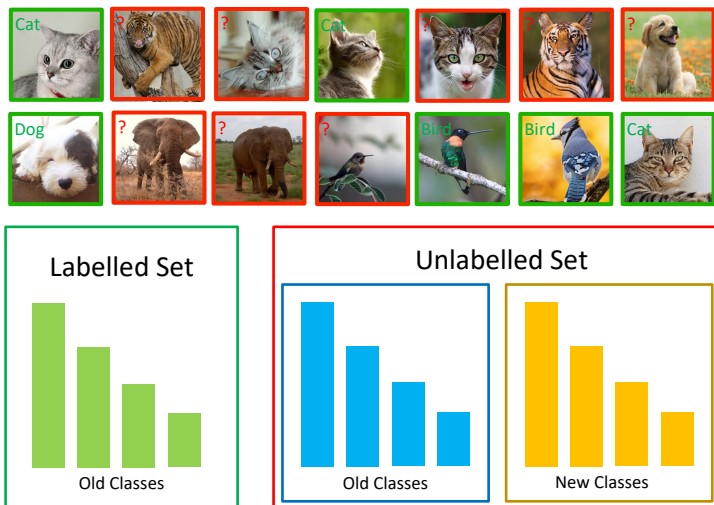

Figure 1: Generalized Category Discovery (GCD) under a long-tailed distribution: Given labelled images from seen categories and unlabelled images from both seen and unseen categories, the objective is to automatically assign labels to the unlabelled images, where the distribution in the labelled and unlabelled data is long-tailed.

the model with a prior distribution of confident samples, further mitigating bias. This approach ensures that the model learns an unbiased representation, enabling accurate recognition of both head and tail classes.

Second, we address the challenge of accurately estimating the number of classes in the long-tailed setting. Existing methods relying on the k-means algorithm have shown to be inaccurate for this purpose. Therefore, we propose a unique density-peak-based class number estimation method that is insensitive to imbalances in the data distribution. By leveraging feature density peaks in the unlabelled data, our method provides a more precise estimation of the number of classes, enhancing the overall performance of the generalized category discovery task.

Experimental results on both long-tailed and conventional GCD datasets demonstrate the effectiveness of our proposed method. Compared to state-of-the-art approaches, our framework achieves superior performance in discovering novel categories accurately and handling the challenges posed by long-tailed distributions. Our contributions extend the scope of generalized category discovery research and offer promising possibilities for real-world applications in computer vision tasks.

## 2 Related Works

Our work is related to the fields of category discovery, semi-supervised learning, and long-tailed distribution recognition, we briefly review the related works below.

*Category Discovery* (CD) was initially introduced as the problem of Novel Category Discovery (NCD) Han et al. (2019), which aims to discover novel categories in the unlabelled assuming disjoint class spaces between labelled and unlabelled data. Building upon NCD, the problem was further developed into a broader framework known as Generalized Category Discovery (GCD) Vaze et al. (2022a). Unlike NCD, GCD relaxes the assumption and permits the unlabelled dataset to include instances from the seen (labelled) classes. For NCD, several approaches have been proposed, such as RankStat RankStat (Han et al., 2021) shows that the NCD task benefits from self-supervised pretrained objectives and proposed a pair-wise objective to transfer the knowledge from the labelled set to the unlabelled dataset. DualRank (Zhao & Han, 2021) extends on this method to use local fine-grained image features and achieves a better performance on fine-grained datasets. Contrastive Learning (Zhong et al., 2021a; Jia et al., 2021) and data augmentations (Zhong et al., 2021b) have also been explored by previous works. UNO (Fini et al., 2021) proposes a unified objective to optimize for the NCD task and obtained state-of-the-art performance. Regarding GCD, (Vaze et al., 2022a) is the first work that formally introduces the GCD task, combining contrastive learning and semi-supervised

$k$-means, (Vaze et al., 2022a) can learn an effective representation on the unlabelled dataset as well as estimating an accurate class number for the whole unlabelled dataset. Concurrent work ORCA (Cao et al., 2022) also tackles a similar setting as GCD, termed open-world semi-supervised learning. XCon (Fei et al., 2022) improves upon (Vaze et al., 2022a) by introducing the technique of splitting the training dataset into $k$-subgroups. (Wen et al., 2022) proposed a simple baseline for the GCD task and can achieve an impressive result over prior works using a parametric classifier. CiPR (Hao et al., 2024) proposes a hierarchical GCD method by effectively exploiting pseudo positive samples from unlabelled images. GPC (Zhao et al., 2023) introduces a GCD framework based on the Gaussian Mixture Model by jointly considering the representation learning and category estimation. Despite the progress, most prior works assume the distribution of categories in the labelled and unlabelled dataset is uniform, which is not representative of real-world data that typically exhibits a long-tailed distribution. NCDLR (Chuyu et al., 2023) considers the problem of NCD under the long-tailed distribution and introduces an adaptive self-labeling method to tackle the challenge. However, NCDLR (Chuyu et al., 2023) limits the study on the NCD setting where the unlabelled dataset is assumed to contain no overlapping classes with the labelled dataset. Recent works (Li et al., 2023b;a) have extended the study of long-tailed distribution to the GCD setting and have proposed methods based on optimal transport (Li et al., 2023a) and reweighting (Li et al., 2023b) to improve the category discovery performance. In this work, we also aim to tackle this realistic long-tailed setting for GCD. As will be shown in the experiments, our proposed method achieves notable improvements over existing methods.

*Semi-Supervised Learning* (SSL) is a long-standing problem that has many effective methods proposed (Rebuffi et al., 2020; Sohn et al., 2020; Berthelot et al., 2019; Laine & Aila, 2017; Tarvainen & Valpola, 2017). The main assumption of SSL is that the unlabelled dataset shares the same set of categories with the labelled dataset, and the goal is to learn a classification model that is able to leverage the unlabelled dataset to improve its classification performance. Self-supervised representations that can help learn a strong representation, are also shown to be effective for SSL (Rebuffi et al., 2020; Zhai et al., 2019). Consistency methods are among the most effective methods for SSL, such as Mean-Teacher (Tarvainen & Valpola, 2017), MixMatch (Berthelot et al., 2019), and FixMatch (Sohn et al., 2020). Recent works shift the attention to a more realistic scenario where the assumption is that the unlabelled dataset can contain categories that are not in the labelled datasets (Saito et al., 2021; Yu et al., 2020; Huang et al., 2021), this setting is termed as open-set SSL. The main difference between open-set SSL and the GCD setting tackled in this paper is that open-set SSL simply rejects unlabelled images from unseen categories, while GCD categorizes all the unlabelled images.

*Long-tailed Recognition* is a long-standing problem which aims at tackling the naturally occurring long-tailed distribution in real-world datasets, where a few classes contain numerous examples (head classes) but other classes only have a few instances (tail classes). The major technical challenge in this setting is that the trained model is easily biased towards head classes and performs poorly on the tail classes. Existing works in long-tailed distribution often assume a fully-supervised setting, several techniques have been proposed, such as re-sampling (Chawla et al., 2002; Guo & Wang, 2021; Kang et al., 2019; Hong et al., 2021), re-weighting (Cao et al., 2019; Deng et al., 2021; He et al., 2022), logits adjustment (Menon et al., 2020; Peng et al., 2022; Tian et al., 2021) and ensembling (Zhou et al., 2020; Wang et al., 2021). Few works focus on the long-tailed semi-supervised learning scenario, and it has been shown that similar techniques like re-sampling or re-weighting (He et al., 2021; Wei et al., 2021; Oh et al., 2022; Lai et al., 2022; Guo & Li, 2022) still work under the semi-supervised setting. However, these long-tailed SSL works still follow the assumption in common SSL scenarios where the unlabelled dataset contains the same set of categories as the labelled set, *i.e.*, no novel categories in the unlabelled dataset. In this work, we consider the case where not only the distribution of the dataset is long-tailed, but also there may exists novel categories in the unlabelled dataset.

# 3    Method

## 3.1    Preliminaries

### 3.1.1    Problem Setting

Generalized Category Discovery (GCD) aims to learn a model for categorizing unlabelled samples in dataset $\mathcal{D}^u = \{(\boldsymbol{x}_i^u, \boldsymbol{y}_i^u)\} \in \mathcal{X} \times \mathcal{Y}_u$, using the knowledge obtained from labelled dataset $\mathcal{D}^l = \{(\boldsymbol{x}_i^l, \boldsymbol{y}_i^l)\} \in \mathcal{X} \times \mathcal{Y}_l$. $\mathcal{D}^u$ consists of unlabelled examples in label space $\mathcal{Y}_u$, while $\mathcal{D}^l$ contains labelled examples in label space $\mathcal{Y}_l$, where $\mathcal{Y}_l \subset \mathcal{Y}_u$. The number of categories in $\mathcal{Y}_u$ is denoted by $K_u$, which is typically assumed to be known a priori or can be estimated using previous methods (Vaze et al., 2022a; Han et al., 2019). Unlike Semi-Supervised Learning (SSL) and Novel Category Discovery (NCD) settings, where $\mathcal{Y}_l = \mathcal{Y}_u$ and $\mathcal{Y}_l \cap \mathcal{Y}_u = \emptyset$, respectively, GCD is a more realistic and practical problem. In our paper, we consider the realistic setting in which the unlabelled data exhibits a long-tailed distribution. Formally, we denote the number of examples in class $k$ as $N_k$, thus $\sum_{k=1}^{K_u} N_k = N$ where $N$ is the number of all examples. Without the loss of generality, the classes are sorted by $N_k$ in descending order ($N_1 \geq N_2 \geq \cdots \geq N_k$), and we can therefore represent the imbalance ratio as $\lambda = \frac{N_1}{N_k}$.

Current approaches for addressing the generalized category discovery problem typically involve two main components: representation learning and label assignment. The label assignment methods can be further subdivided into two distinct categories - parametric classifiers and non-parametric clustering methods.

In the following sections, we will begin by presenting a parametric classification baseline (Sec. 3.1.2). Subsequently, we will introduce our proposed method for handling long-tailed distributions, which encompasses a sample selection process optimized for achieving balance during classifier training (Sec. 3.2). Furthermore, we will delve into a density-based class number estimation module capable of accurately estimating the class number under long-tailed distributions (Sec. 3.3).

### 3.1.2    Baseline

We first present the strong GCD baseline proposed in (Wen et al., 2022) which contains two parts, representation learning and classifier learning.

**Representation learning** aims to learn a general representation of all classes that can be further utilized by the classifier to classify images from both labelled and unlabelled classes. The representation learning utilizes supervised contrastive learning (Khosla et al., 2020) for labelled data and self-supervised contrastive learning (Chen et al., 2020) for all the data. The overall representation learning loss :

$$\mathcal{L}_{\mathrm{rep}} = \lambda_{\mathrm{rep}} \mathcal{L}_{\mathrm{SupCon}} + (1 - \lambda_{\mathrm{rep}}) \mathcal{L}_{\mathrm{SelfCon}}, \tag{1}$$

where $\mathcal{L}_{\mathrm{SupCon}}$ and $\mathcal{L}_{\mathrm{SelfCon}}$ denote supervised contrastive loss and self-supervised contrastive loss respectively. $\lambda_{\mathrm{rep}}$ is a balancing factor.

**Classifier learning** aims to learn a classifier for all the classes in the dataset based on the learned representations. We can define a set of prototypes $\mathcal{C} = \{\boldsymbol{c}_1, \ldots, \boldsymbol{c}_K\}$ where $K_u$ is the total number of classes in the dataset. During training, we first calculate the predicted logits of one augmented view $\hat{\boldsymbol{x}}_i$ of the input $\boldsymbol{x}_i$ belonging to each class $k$ using the hidden features $\hat{\boldsymbol{h}}_i = f(\hat{\boldsymbol{x}}_i)$ with normlization:

$$l^{(k)}(\hat{\boldsymbol{x}}_i) = (\hat{\boldsymbol{h}}_i / \|\hat{\boldsymbol{h}}_i\|_2)^\top (\boldsymbol{c}_k / \|\boldsymbol{c}_k\|_2) / \tau_s. \tag{2}$$

We then use the softmax function to convert these logits to a probability:

$$\hat{\boldsymbol{p}}_i^{(k)} = p^{(k)}(\hat{\boldsymbol{x}}_i) = \frac{\exp l^{(k)}}{\sum_{k'} \exp l^{(k')}}. \tag{3}$$

We use the other view $\tilde{\boldsymbol{x}}_i$ of the same input $\boldsymbol{x}_i$ to calculate the soft pseudo-label $\tilde{\boldsymbol{p}}_i$ with a sharpen temperature $\tau_t$: $l^{(k)}(\tilde{\boldsymbol{x}}_i) = (\tilde{\boldsymbol{h}}_i / \|\tilde{\boldsymbol{h}}_i\|_2)^\top (\boldsymbol{c}_k / \|\boldsymbol{c}_k\|_2) / \tau_t$. Next, we follow (Wen et al., 2022) to train the classifier with the

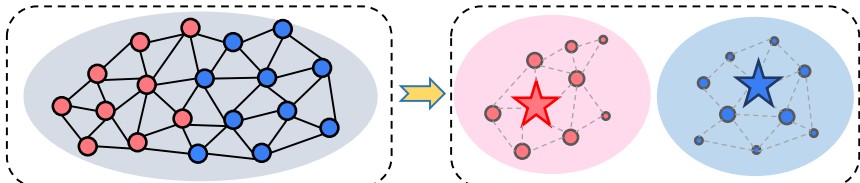

Figure 2: Density selection process. Left: we compute the similarity between the nearest neighbors of each data sample, denoted as the edges in the figure. Right: leveraging the density definition in Eq. (7), we select a few density peaks from the raw data.

weighted sum of the following two losses:

$$\mathcal{L}_{\text{cls}}^u = \frac{1}{|B|} \sum_{i \in B} \ell(\tilde{\boldsymbol{p}}_i, \hat{\boldsymbol{p}}_i) - \epsilon H(\overline{\boldsymbol{p}}), \mathcal{L}_{\text{cls}}^s = \frac{1}{|B^l|} \sum_{i \in B^l} \ell(\boldsymbol{y}_i, \hat{\boldsymbol{p}}_i), \tag{4}$$

where $\ell(\boldsymbol{q}, \boldsymbol{p}) = -\sum_k \boldsymbol{q}^{(k)} \log \boldsymbol{p}^{(k)}$ is the cross-entropy loss, $\boldsymbol{y}_i$ is the label of $\boldsymbol{x}_i$, and $H(\overline{\boldsymbol{p}})$ is an entropy regularization as adopted (Assran et al., 2022). Particularly, $H(\overline{\boldsymbol{p}}) = -\sum_k \overline{\boldsymbol{p}}^{(k)} \log \overline{\boldsymbol{p}}^{(k)}$, where $\overline{\boldsymbol{p}}$ is the mean prediction of the batch, computed by $\overline{\boldsymbol{p}} = \frac{1}{2|B|} \sum_{i \in B} (\hat{\boldsymbol{p}}_i + \tilde{\boldsymbol{p}}_i)$. The classification loss is then written as $\mathcal{L}_{\text{cls}} = (1 - \lambda_{\text{cls}})\mathcal{L}_{\text{cls}}^u + \lambda_{\text{cls}}\mathcal{L}_{\text{cls}}^s$. The overall loss for representation learning and classifier learning losses is then written as: $\mathcal{L}_{\text{base}} = \mathcal{L}_{\text{rep}} + \mathcal{L}_{\text{cls}}$.

## 3.2 GCD Classifier for Long-tailed Distribution

One major challenge that arises from the long-tailed data distribution is that the classifier may be biased towards the head classes, which have much more data than the tail classes. This could result in unreliable pseudo-labels for training and, thus, hurt the learned representation and generalization. To address the challenge of training a model with a long-tailed data distribution, we propose leveraging a sample selection method to curate a balanced subset containing reliable samples from the unlabelled dataset. During training, we will exclusively use this subset of examples to construct training mini-batches and enforce the model's prediction distribution to closely match the distribution of the selected subset. The underlying intuition behind this approach is based on selecting a subset of high-quality data that exhibits a relatively balanced distribution. By doing so, we aim to assist the model in mitigating the bias originating from the original long-tailed distribution.

We introduce two complementary methods for selecting reliable samples in the unlabelled dataset, one is based on the prediction confidence of the input example $\boldsymbol{x}_i$ (relying on only the individual sample), and the other one is based on using the density of each data samples (Xing et al., 2021) (relying on the neighbors of a sample). Formally, for the confidence-based selection, we use the prototype classifier $p(\cdot)$ introduced in Sec. 3.1.2 with the sharpened temperature $\tau_t$ to obtain the prediction of the model for each sample in the unlabelled dataset $\boldsymbol{p}_i = p(\boldsymbol{x}_i), \boldsymbol{x}_i \in \mathcal{D}^u$. With this prediction, we sample a subset of the unlabelled data example $\mathcal{S}_{\text{conf}}$ using:

$$\mathcal{S}_{\text{conf}} = \{\boldsymbol{x}_i | p(\boldsymbol{x}_i) \geq \epsilon_{\text{conf}}, \boldsymbol{x}_i \in \mathcal{D}^u\}, \tag{5}$$

where $\epsilon_{\text{conf}}$ denotes the threshold for the confidence selection. For the density-based sample selection, we adopted the density definition from (Xing et al., 2021) which defines the density $d_i$ of a sample $\boldsymbol{x}_i$ as:

$$d_i = \frac{1}{|\mathcal{N}_{\boldsymbol{x}_i}^k|} \sum_{j \in \mathcal{N}_{\boldsymbol{x}_i}^k} e_{ij} \cdot a_{ij}, \tag{6}$$

where $\mathcal{N}_{\boldsymbol{x}_i}^k$ is the set of $k$ nearest neighbor of the sample $\boldsymbol{x}_i$. To calculate the connectivity $e_{ij}$ between sample $\boldsymbol{x}_i$ and its $j$-th neighbor, we use the formula $e_{ij} = 2\boldsymbol{p}_i \cdot \boldsymbol{p}_j - 1$. Additionally, we determine the affinity $a_{ij}$ between $\boldsymbol{x}_i$ and $\boldsymbol{x}_j$ as $a_{ij} = <\boldsymbol{h}_i, \boldsymbol{h}_j>$. Note that the choice of the density definition is not unique, and alternative density estimation methods can also be applied in this context. Intuitively, the density $d_i$ reflects the compactness of the embedding space around a sample $\boldsymbol{x}_i$, where a higher density implies that $\boldsymbol{x}_i$ is closer to the center of its corresponding class. Importantly, this density measurement is independent of the quality

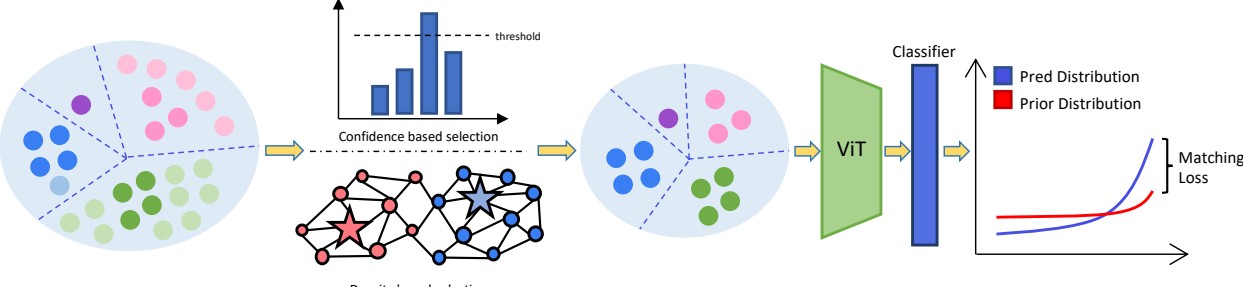

Figure 3: The overall framework of classifier training. From the original long-tailed dataset, we leverage two complementary methods to select a reliable subset of the data to form training mini-batches. The model is trained with the subsampled data, as well as being regularized by a matching loss to produce a prediction distribution similar to the selected dataset to reduce biases caused by the long-tailed distribution.

of the learned classifier since the comparison is solely conducted within the embedding space. Consequently, we can consider this property as being irrelevant to the classifier's performance, as the selection process only considers the embedding space and is not heavily influenced by the long-tailed distribution. With the density estimation for each sample, we propose to select a set of density peaks $\mathcal{S}_{\mathrm{dens}}$ from the unlabelled dataset by:

$$\mathcal{S}_{\mathrm{dens}} = \mathtt{NMDS}(\{\boldsymbol{x}_i | \forall j \in \mathcal{N}_{\boldsymbol{x}_i}^k, d_i \geq d_j, \boldsymbol{x}_i \in \mathcal{D}^u\}). \tag{7}$$

Here, we first identify a set of density peaks which consist of samples with higher density than their $k$ nearest neighbors. We then use the $\mathtt{NMDS}$ function for Non-Maximum Density Suppression, defined in Algorithm 1, to suppress redundant high-density samples in the head class. In this algorithm, the intersection-over-union function is defined as $\mathtt{IoUK}(\boldsymbol{x}_i, \boldsymbol{x}_j) = \frac{\|\mathcal{N}_{\boldsymbol{x}_i}^{k_s} \cap \mathcal{N}_{\boldsymbol{x}_j}^{k_s}\|}{\|\mathcal{N}_{\boldsymbol{x}_i}^{k_s} \cup \mathcal{N}_{\boldsymbol{x}_j}^{k_s}\|}$, where $k_s$ is a hyper-parameter that sets the number of neighbors to compare in the function. Ultimately, this process allows us to obtain $\mathcal{S}_{\mathrm{dens}}$, a subset of the unlabelled dataset consisting of density peaks. We provide a visualization in Fig. 2 to explain the density selection process.

Together with the confidence selection, the final selected data samples form $\mathcal{S} = \mathcal{S}_{\mathrm{conf}} \cup \mathcal{S}_{\mathrm{dens}}$. The prior distribution $\boldsymbol{p}_{\mathrm{prior}}$ is formed using the pseudo-label distribution within this selected subset, as reliable samples can provide a more balanced distribution for the model to learn, specifically:

$$\boldsymbol{p}_{\mathrm{prior}} = \sigma(\sum_{\boldsymbol{x} \in \mathcal{S}} \hat{y}(\boldsymbol{x})), \tag{8}$$

where $\sigma$ is the softmax function, and $\hat{y}$ is a function that generate a one-hot pseudo label of the input $\boldsymbol{x}$ using the prediction $\boldsymbol{p}$. An additional loss is then added to the model:

$$\mathcal{L}_{\mathrm{match}} = \ell(\boldsymbol{p}_{\mathrm{prior}}, \overline{\boldsymbol{p}}), \tag{9}$$

Here, $\ell$ is the cross-entropy function and $\overline{\boldsymbol{p}}$ represents the target distribution. This regularizer will drive the model to match its predicted distribution with the selected reliable sample distribution, thus improving the classifier and the underlying representations. The overall loss of the model is $\mathcal{L} = \mathcal{L}_{\mathrm{base}} + \mathcal{L}_{\mathrm{match}}$.

Additionally, we apply this sample selection method to the whole unlabelled dataset $\mathcal{D}^u$, selecting a subset $\hat{\mathcal{D}}^u = \mathcal{S}$ at the end of each epoch based on the same criteria used for the labelled dataset. At the start of the next epoch, we draw unlabelled training mini-batches $B$ only from the subset $\hat{\mathcal{D}}^u$. We illustrate the overall framework of our proposed method for training the classifier in Fig. 3.

## 3.3 Class Number Estimation

Another major challenge with the long-tailed distribution in generalized category discovery is that it can be hard to estimate the number of classes in the unlabelled set using the conventional semi-supervised $k$-means (Vaze et al., 2022a) algorithm, as the $k$-means algorithm assumes that each of the clusters in the data

**Algorithm 1** Non-Maximum Density Suppression

```
procedure NMDS(S)
    S_NMDS ← ∅
    for (x_i, d_i) ∈ S do
        discard ← False
        for (x_j, d_j) ∈ S do
            if IoUK(x_i, x_j) > λ_NMDS then
                if d_i > d_j then
                    discard ← True
        if not discard then
            S_NMDS ← S_NMDS ∪ (x_i, d_i)
    return S_NMDS
```

Table 1: Statistics of the datasets.

| | Labelled | | Unlabelled | | |
| Dataset | #Image | #Class | #Image | #Class | $\lambda$ |
|---|---|---|---|---|---|
| CUB | 1.5K | 100 | 4.5K | 200 | 1.0 |
| Stanford Cars | 2.0K | 98 | 6.1K | 196 | 1.0 |
| ImageNet-100 | 31.9K | 50 | 95.3K | 100 | 1.0 |
| CIFAR-100-LT | 4K | 80 | 11K | 100 | 10.0 |
| ImageNet-100-LT | 6K | 50 | 22K | 100 | 10.0 |
| Herbarium 19 | 9K | 341 | 25K | 683 | 46.1 |
| iNatualist-18 | 130K | 4,071 | 307K | 8,142 | 500.0 |

is isotropic with roughly equal number of samples per cluster (Liang et al., 2012; Wu et al., 2009). This assumption, however, does not hold in the long-tailed distribution we want to tackle, thus we cannot directly adopt the semi-supervised $k$-means algorithm for estimating the number of classes in the long-tailed GCD setting. Here we propose a novel algorithm for determining the number of categories in the unlabelled set using the concept of the $k$-NN density of data samples (Xing et al., 2021). The main idea is akin to density-based selection, where samples with higher density and density peaks are more likely to be in proximity to the cluster center of a cluster (Ester et al., 1996; Xing et al., 2021). By quantifying the number of density peak samples in the unlabelled data, we can estimate the potential number of classes present in the dataset.

Formally, given a dataset of samples $\mathcal{D}^u = \{(\boldsymbol{x}_i^u, \boldsymbol{y}_i^u)\}$, our goal is to determine the number of classes $K_u$ in $\mathcal{D}^u$ as well as an assignment $\hat{\boldsymbol{y}}_i$ of each data samples $\boldsymbol{x}_i^u$. Rather than employing iterative calculations of class prototypes and updating class assignments, our approach leverages the idea of a 'density peak' in the dataset (Ester et al., 1996; Xing et al., 2021) to develop an algorithm for estimating the value of $K_u$. A 'density peak' $\boldsymbol{x}_i$ is defined as the density $d_i$ is higher than all the $k$ neighbours of $\boldsymbol{x}_i$, $\forall j \in \mathcal{N}_{\boldsymbol{x}_i}, d_i > d_j$. After getting all the density peaks in the dataset $\mathcal{D}^u$, we run an NMDS step using the algorithm defined in Algorithm 1 to remove duplicated density peaks that may belong to the same categories. he class assignment process involves utilizing the density peaks as class prototypes and employing a straightforward distance-based assignment method. To determine the number of classes, we establish an upper bound by considering the total count of density peaks, while the lower bound corresponds to the number of labelled categories. Employing Brent's algorithm (Press et al., 1992) in conjunction with our density-based approach on the combined labelled and unlabelled dataset, we search for the optimal new category number. Throughout this process, we discard the labels associated with the labelled data. The optimal value is determined by evaluating the clustering accuracy on the labelled data, aiming to maximize its performance.

## 4 Experiments

### 4.1 Experimental Setup

**Benchmark and evaluation metrics.** We validate the performance of our methods on long-tailed datasets, including long-tailed CIFAR-100 (Krizhevsky & Hinton, 2009), ImageNet-100 (Deng et al., 2009) as well as naturally occuring long-tailed datasets including Herbarium-19 (Tan et al., 2019) and iNat-18 (Van Horn et al., 2018). We also conduct experiments on uniformed distributed datasets for GCD, including ImageNet-100 (Deng et al., 2009) and the Semantic Shift Benchmark(SSB) (Vaze et al., 2022b) which includes CUB (Wah et al., 2010) and Stanford Cars (Krause et al., 2013). For each of the datasets, we follow previous works (Vaze et al., 2022a; Wen et al., 2022) to sample a subset of all classes as the old classses $\mathcal{Y}_l$; 50% of the images from these labelled classes are used to construct $\mathcal{D}^l$, and the remaining images are regarded as the unlabelled data $\mathcal{D}^u$. Please refer to Tab. 1 for statistics of the datasets we evaluate on, as well as the imbalance factor $\lambda$ of each datasets. The model is evaluated using the clustering accuracy

(ACC) following the standard practice (Vaze et al., 2022b; Wen et al., 2022). For the long-tailed dataset, we compute the balanced-ACC as the average of per-class ACC for an unbiased evaluation.

Table 2: Results on datasets with a long-tailed distribution.

| No. Methods | CIFAR100-LT | | | ImageNet100-LT | | | Herb-19 | | |
|---|---|---|---|---|---|---|---|---|---|
| | All | Old | New | All | Old | New | All | Old | New |
| (1) $k$-means (MacQueen, 1967) | 31.3 | 35.3 | 30.1 | 51.9 | 67.2 | 30.8 | 13.6 | 12.2 | 15.0 |
| (2) RankStats+ (Han et al., 2021) | 45.7 | 59.1 | 24.1 | 47.4 | 70.1 | 23.1 | 11.4 | 13.2 | 12.5 |
| (3) UNO+ (Fini et al., 2021) | 49.9 | 61.1 | 25.7 | 51.2 | 74.2 | 25.1 | 15.3 | 17.1 | 13.4 |
| (4) ORCA (Cao et al., 2022) | 41.3 | 59.5 | 20.7 | 46.3 | 67.1 | 24.2 | 9.8 | 14.7 | 4.9 |
| (5) GCD (Vaze et al., 2022a) | 62.3 | 66.9 | 28.1 | 53.1 | 75.1 | 28.3 | 32.8 | 41.4 | 24.2 |
| (6) SimGCD (Wen et al., 2022) | 70.4 | 77.4 | 32.1 | 56.6 | 79.6 | 33.5 | 39.4 | 51.4 | 27.3 |
| (7) NCDLR (Chuyu et al., 2023) | 67.8 | 73.4 | 31.5 | 55.7 | 74.5 | 33.7 | 38.6 | 48.7 | 28.0 |
| (8) ImbaGCD (Li et al., 2023a) | 71.5 | **81.2** | 33.0 | 57.4 | 81.2 | 34.8 | 42.5 | 54.7 | **28.9** |
| (9) Ours | **72.1**$_{\pm0.4}$ | 80.1$_{\pm1.2}$ | **33.7**$_{\pm0.8}$ | **58.4**$_{\pm0.2}$ | **83.1**$_{\pm0.5}$ | **35.4**$_{\pm1.5}$ | **43.5**$_{\pm0.9}$ | **55.8**$_{\pm0.6}$ | 28.5$_{\pm0.4}$ |

**Implementation details.** Following the common practice, we train all methods with the ViT-base/16 model (Dosovitskiy et al., 2021) pretrained with DINO (Caron et al., 2021). The `[CLS]` token with a dimension of 768 is used as the feature representation of one image and we only finetune the last block of the backbone. The model is trained with a batch-size of 128, with an initial learning rate of 0.1 decayed with a cosine schedule to 0. $\epsilon_{\text{conf}}$ is set to 0.8. For a fair comparison, we train for 200 epochs on each dataset, and the best-performing model is selected using the accuracy of the validation set of the labelled classes.

## 4.2 Comparison with the State-of-the-Art

We present a comparison of our method with state-of-the-art methods on both long-tailed datasets (see Tab. 2) and the SSB benchmark datasets (see Tab. 3). From Tab. 2 we can observe that our method achieves the overall best performance on the challenging long-tailed distribution datasets, outperforming the second-best model ImbaGCD (Li et al., 2023a) by 0.6%-1.9% in ACC, validating the effectiveness of our method for handling the long-tailed distribution. It can also be observed that our method demonstrates a non-trivial performance improvement over them compared with other previous state-of-the-art. When comparing with alternative methods designed to tackle long-tailed distributions, our proposed method is the only method that consistently outperforms the strong baseline SimGCD (Wen et al., 2022) in terms of the performance on the long-tailed 'New' categories. This observation highlights the challenges associated with handling long-tailed distributions in category discovery, and also demonstrates the potential of our proposed method in handling this real-world long-tailed challenge. As shown in Table 2, our method achieves the highest performance across most scenarios, demonstrating its superiority. In Tab. 3, our method demonstrates superior performance on both the 'All' and 'New' classes, while maintaining comparable performance to SimGCD on the "old" classes. It is worth noting that our method is specifically designed to address the challenges posed by long-tailed scenarios. Therefore, achieving performance similar to strong methods on conventional benchmarks is an encouraging outcome, indicating the versatility and potential applicability of our approach across different scenarios.

## 4.3 Novel Class Number Estimation

In Tab. 4, we show the performance of estimating the number of categories in the unlabeled dataset. We first show a comparison of estimated category numbers on uniformed datasets including CIFAR-100 and ImageNet-100, compared with the search algorithm proposed in (Vaze et al., 2022a), our method gives comparable estimation performance. Importantly, our method showcases substantial improvements in estimating the real-world long-tailed distribution datasets, encompassing both artificially split long-tailed datasets such as CIFAR-100-LT and ImageNet-100-LT, as well as naturally occurring long-tailed datasets like Herb-19 and iNat-18. Notably, our method outper-

Table 4: Estimation of class numbers in unlabelled data.

| Dataset | GT | GCD | Ours |
|---|---|---|---|
| CIFAR100 | 100 | **100** | 109 |
| ImageNet-100 | 100 | **109** | 112 |
| CIFAR100-LT | 100 | 78 | **86** |
| ImageNet-100-LT | 100 | 71 | **79** |
| Herb-19 | 683 | 520 | **586** |
| iNat-18 | 8,142 | 5981 | **6,151** |

Table 3: Results on the Semantic Shift Benchmark (Vaze et al., 2022b).

| | | CUB | | | Stanford Cars | | | ImageNet-100 | | |
|---|---|---|---|---|---|---|---|---|---|---|
| No. | Methods | All | Old | New | All | Old | New | All | Old | New |
| (1) | $k$-means (MacQueen, 1967) | 34.3 | 38.9 | 32.1 | 12.8 | 10.6 | 13.8 | 72.7 | 75.5 | 71.3 |
| (2) | RankStats+ (Han et al., 2021) | 33.3 | 51.6 | 24.2 | 28.3 | 61.8 | 12.1 | 37.1 | 61.6 | 24.8 |
| (3) | UNO+ (Fini et al., 2021) | 35.1 | 49.0 | 28.1 | 35.5 | 70.5 | 18.6 | 70.3 | **95.0** | 57.9 |
| (4) | ORCA (Cao et al., 2022) | 35.3 | 45.6 | 30.2 | 23.5 | 50.1 | 10.7 | 73.5 | 92.6 | 63.9 |
| (5) | GCD (Vaze et al., 2022a) | 51.3 | 56.6 | 48.7 | 39.0 | 57.6 | 29.9 | 74.1 | 89.8 | 66.3 |
| (6) | SimGCD (Wen et al., 2022) | 60.3 | **65.6** | 57.7 | 46.8 | **64.9** | 38.0 | 82.4 | 90.7 | 78.3 |
| (7) | NCDLR (Chuyu et al., 2023) | 58.7 | 58.7 | 52.4 | 44.5 | 60.1 | 34.5 | 77.5 | 89.9 | 74.5 |
| (8) | ImbaGCD (Li et al., 2023a) | 61.0 | 64.0 | 58.4 | 47.4 | 63.2 | 39.0 | **82.5** | 90.8 | **78.9** |
| (9) | Ours | $61.3_{\pm 0.1}$ | $64.2_{\pm 0.9}$ | $59.2_{\pm 0.4}$ | $47.9_{\pm 1.8}$ | $64.7_{\pm 1.3}$ | $39.3_{\pm 2.1}$ | $81.1_{\pm 2.2}$ | $88.4_{\pm 2.2}$ | $77.8_{\pm 2.7}$ |

Table 5: Evaluation with different imbalance factors $\lambda$. All results are in 'All / Old / New'.

| CIFAR-100-LT | SimGCD | Ours |
|---|---|---|
| $\lambda = 5$ | 73.5 / 80.2 / 35.1 | 74.2 / 81.0 / 35.6 |
| $\lambda = 10$ | 70.4 / 77.4 / 32.1 | 72.1 / 80.1 / 33.7 |
| $\lambda = 20$ | 63.1 / 70.3 / 28.6 | 67.2 / 75.3 / 30.2 |
| ImageNet-100-LT | SimGCD | Ours |
| $\lambda = 5$ | 62.1 / 83.1 / 37.1 | 63.1 / 84.5 / 38.1 |
| $\lambda = 10$ | 56.6 / 79.6 / 33.5 | 58.4 / 83.1 / 35.4 |
| $\lambda = 20$ | 50.1 / 74.5 / 26.3 | 54.2 / 78.2 / 29.1 |

Table 6: Class number estimation with different $\lambda$.

| CIFAR-100-LT | GCD | Ours |
|---|---|---|
| $\lambda = 5$ | 87 | **89** |
| $\lambda = 10$ | 78 | **86** |
| $\lambda = 20$ | 65 | **80** |
| ImageNet-100-LT | GCD | Ours |
| $\lambda = 5$ | 85 | **87** |
| $\lambda = 10$ | 71 | **79** |
| $\lambda = 20$ | 60 | **73** |

forms the approach proposed in (Vaze et al., 2022a) in these contexts. These results serve as compelling evidence for the effectiveness of our method when applied to long-tailed datasets.

## 4.4 Ablation Study

In this section, we provide ablations to each component of our method.

**Performance with different imbalance factors $\lambda$.** Firstly, we present an ablation to study the performance variation when the imbalance factor $\lambda$ is different. We create splits with different $\lambda$ values by subsampling CIFAR-100 and ImageNet-100 datasets. The clustering results, as shown in Tab. 5, consistently demonstrate the superior performance of our method across the different $\lambda$ values tested. Notably, we observe that as $\lambda$ increases, the performance gap between our method and SimGCD widens. This finding emphasizes the effectiveness of our proposed method in handling the long-tailed distribution.

The results of estimating the category numbers are presented in Tab. 6, which reveal that as $\lambda$ increases, the estimated number decreases due to the fact that the smallest cluster becomes smaller with higher imbalance, and it is more likely for the estimation algorithm to overlook it. According to the results, our proposed method outperforms the algorithm in (Vaze et al., 2022a) in all scenarios.

**Confident sample selection.** In Tab. 8, we show an ablation study using different combinations of the selected subset $\mathcal{S}$. The default choice is to use $\mathcal{S}_{\mathrm{conf}} \cup \mathcal{S}_{\mathrm{dens}}$ as $\mathcal{S}$ for forming training mini-batches to train the model. Here we explore the performance of only using $\mathcal{S}_{\mathrm{conf}}$ or $\mathcal{S}_{\mathrm{dens}}$ as $\mathcal{S}$. From Tab. 8, we can observe that removing $\mathcal{S}_{\mathrm{conf}}$ or $\mathcal{S}_{\mathrm{dens}}$ results in a performance degradation. The performance degrades more when $\mathcal{S}_{\mathrm{conf}}$ is removed from $\mathcal{S}$, the gap is about 10% on 'All', 'Old', and 'New' categories. These results demonstrate that both $\mathcal{S}_{\mathrm{conf}}$ and $\mathcal{S}_{\mathrm{dens}}$ are essential to the final performance of the model validating the design choice of our method.

Table 7: Imbalance factor $\lambda$ within the selected subset $\mathcal{S}$ Table 8: Results with different selected samples $\mathcal{S}$.

|  | ImageNet-100-LT | Herb-19 |
|---|---|---|
| $\mathcal{D}^u$ | 10.0 | 46.1 |
| $\mathcal{S}_{\mathrm{conf}}$ | 8.4 | 44.3 |
| $\mathcal{S}_{\mathrm{dens}}$ | **2.0** | **10.2** |
| $\mathcal{S}_{\mathrm{conf}} \cup \mathcal{S}_{\mathrm{dens}}$ | 4.5 | 20.7 |

|  | ImageNet-100-LT | Herb-19 |
|---|---|---|
| Ours w/o $\mathcal{S}_{\mathrm{conf}}$ | 40.2 / 71.2 / 22.5 | 30.0 / 38.5 / 10.4 |
| Ours w/o $\mathcal{S}_{\mathrm{dens}}$ | 55.1 / 80.2 / 31.4 | 40.2 / 51.4 / 24.6 |
| Ours | **58.4 / 83.1 / 35.4** | **43.5 / 55.8 / 28.5** |

Table 9: Effectiveness of the `NMDS` algorithm.

|  | ImageNet-100-LT | Herb-19 |
|---|---|---|
| Ours w/o `NMDS` | 55.1 / 76.2 / 31.8 | 40.1 / 50.7 / 25.1 |
| Ours w/ `NMDS` | 58.4 / 83.1 / 35.4 | 43.5 / 55.8 / 28.5 |
| $\lambda_{\mathcal{S}}$ w/o `NMDS` | 7.8 | 36.5 |
| $\lambda_{\mathcal{S}}$ w/ `NMDS` | 4.5 | 20.7 |

**How balanced is the selected subset?** We show the imbalance factor $\lambda$ of the distribution within the selected subset $\mathcal{S}$. These statistics are shown in Tab. 7. The original imbalance factor $\lambda$ of the whole unlabeled dataset is shown in the first row. We can observe from the following rows that using only the confidence-based selection to select samples for $\mathcal{S}_{\mathrm{conf}}$ is not able to reduce the imbalanced distribution of the dataset. Using the density-based selection can indeed sample a more balanced subset from the original dataset, yet as shown in Tab. 8, using $\mathcal{S}_{\mathrm{dens}}$ alone can not achieve good performance for category discovery. Thus we need to combine these two subsets to form $\mathcal{S} = \mathcal{S}_{\mathrm{conf}} \cup \mathcal{S}_{\mathrm{dens}}$ to enjoy the benefit of a more balanced training set and a better performance simultaneously. The combined $\mathcal{S}$ has a more balanced dataset than the original dataset measured by the imbalanced factor.

**Effect of the `NMDS` algorithm.** We validate the effectiveness of our proposed `NMDS` algorithm by removing it and evaluating the performance in Tab. 9. Comparing the first two rows in Tab. 9, we can see that removing the `NMDS` algorithm leads to a performance drop. To investigate this phenomenon further, we show the imbalance factor $\lambda_{\mathcal{S}}$ of the selected subset $\mathcal{S}$ in the bottom two rows of Tab. 9. We can see that without the use of the `NMDS` algorithm, the imbalance factor $\lambda_{\mathcal{S}}$ would be significantly higher than when we use the `NMDS` algorithm.

**Number of Nearest Neighbours.** We use two numbers of nearest neighbours in our proposed method, one is the number $k$ used as the number of nearest neighbours for calculating the density and density peaks, the other one is the number of $k_s$ used in the `IoUK` function to determine the overlap between two different density peaks. We first ablate on the influence of $k$ for the clustering performance, intuitively, a larger $k$ can cover more neighbors, thus providing a more accurate estimation of density peaks. However, by covering a larger neighbourhood, we would expect the selected number of density peaks to drop as it is less likely for one sample to have a higher density than a larger number of neighbors. In the extreme scenario where the neighborhood size $k$ is equal to the total number of samples in the dataset, there would only be one density peak. In Tab. 10, we can observe that the optimal value for $k$ in our experiments is around 10, and this value is consistent across the ImageNet-100-LT and Herb-19 datasets. We set 10 as the default choice in our method, though the sensitivity to different numbers is not high.

In Tab. 11, we show the ablation on $k_s$ evaluting on the performance for estimating the number of categories. We can observe that when the value of $k_s$ is small, the `IoUK` function can only cover a small region in the embedding to perform `NMDS`, thus the method tends to overestimate the number of categories in the dataset. When the value of $k_s$ is larger than the optimal value, the `IoUK` function will overestimate the similarity between two density peaks, thus the method could make more false negative removal of density peaks, leading to an underestimate of the categories. In our experiments, we set $k_s$ to 30 as the default value.

Table 10: Ablation of $k$ for clustering performance.

| $k$ | ImageNet-100-LT | Herb-19 |
|---|---|---|
| 5 | 56.1 / 82.4 / 33.8 | 41.2 / **56.1** / 27.1 |
| 10 | **58.4** / **83.1** / **35.4** | **43.5** / 55.8 / 28.5 |
| 15 | 57.0 / 81.9 / 33.0 | 42.1 / 54.1 / **29.4** |
| 20 | 55.2 / 80.1 / 31.2 | 40.1 / 52.4 / 26.9 |

Table 11: Ablation of $k_s$ for category number estimation performance.

| $k_s$ | ImageNet-100-LT | Herb-19 |
|---|---|---|
| GT | 100 | 683 |
| 10 | 146 | 761 |
| 20 | 135 | 620 |
| 30 | 79 | 586 |
| 40 | 65 | 511 |

## 5  Conclusion

In this paper, we have addressed the challenge of generalized category discovery in long-tailed distributions, a problem that has been underexplored in the literature. We have identified two key technical challenges - balancing the classifier for all categories and accurately estimating the category numbers in the presence of a long tail. To overcome these challenges, we have proposed a novel method based on sample densities. Our approach iteratively selects a balanced and reliable subset from the original unlabelled dataset and estimates category numbers using density-based clustering. Through our experiments on both long-tailed and uniform datasets, we have demonstrated the effectiveness of our method in accurately discovering novel categories.

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

Table A1: Ablation of $\lambda_{\texttt{NMDS}}$ for category number estimation performance.

| $\lambda_{\texttt{NMDS}}$ | ImageNet-100-LT | Herb-19 |
|---|---|---|
| GT | 100 | 683 |
| 0.2 | 167 | 743 |
| 0.4 | 153 | 617 |
| 0.6 | 113 | 594 |
| 0.8 | 85 | 523 |

Table A2: Comparison of different feature density formulations.

| | ImageNet-100-LT | Herb-19 |
|---|---|---|
| Only Affinity | 40.7 / 64.3 / 10.8 | 23.4 / 41.5 / 9.8 |
| Ours | **58.4 / 83.1 / 35.4** | **43.5 / 55.8 / 28.5** |

## A  Additional ablations

Here we present additional ablation studies for our proposed method.

### A.1  Ablation on $\lambda_{\texttt{NMDS}}$

We first present the ablation on the value of the threshold $\lambda_{\texttt{NMDS}}$. The results are shown in Tab. A1. Similar to the ablation of varying $k_s$ in Table 11 of the main paper, setting $\lambda_{\texttt{NMDS}}$ to a lower value will result in an overestimation of the number of categories, and a higher value will result in an underestimation of categories. In our experiments, we set it to 0.6 for simplicity.

### A.2  Ablation on different choices of density calculation

Here, we compare different choices of the feature density formulation. In the main paper, we use both the connectivity $e_{ij}$ and the affinity $a_{ij}$ between each pair of samples, $\boldsymbol{x}_i$ and $\boldsymbol{x}_j$. This definition of feature density is inherited from HiLander (Xing et al., 2021). In this section, we also experiment with an alternative definition of the feature density, which simply averages the affinity values among samples. The density is defined as:

$$d_i = \frac{1}{|\mathcal{N}_{\boldsymbol{x}_i}^k|} \sum_{j \in \mathcal{N}_{\boldsymbol{x}_i}^k} a_{ij}. \tag{10}$$

We present the results in Tab. A2. We can see that if we only use the affinity for calculating the density, the results will be much worse than using connectivity and affinity together. We conjecture this to the fact that our evaluation is under the long-tailed distribution, thus using affinity alone cannot give a good estimation of the density for tail classes and this leads to the degradation in performance.

## B  Combine with Long-tailed Semi-Supervised Learning

Our method tackles the challenge of learning generalized category discovery under the long-tailed distribution. Long-tailed semi-supervised learning is a neighboring problem to ours, which also assumes a long-tailed distribution of classes. However, it is important to note that our setting differs from long-tailed semi-supervised learning. In our case, we encounter novel categories within the unlabelled images that cannot be directly addressed by long-tailed semi-supervised learning methods, as they are not designed to handle novel categories. This distinction highlights the unique challenge we faced in tackling the discovery of novel categories within a long-tailed distribution, requiring the development of a specialized approach to effectively address this scenario. In this section, we propose to combine our method with long-tailed semi-supervised

Table A3: Performance when combined with long-tailed semi-supervised learning method.

|  | ImageNet-100-LT | Herb-19 |
|---|---|---|
| Ours | 58.4 / 83.1 / 35.4 | 43.5 / 55.8 / 28.5 |
| Ours+CReST | 59.4 / 84.6 / 36.7 | 44.6 / 57.4 / 29.4 |

Table A4: Runtime (in seconds) comparison for category number estimation.

|  | ImageNet-100-LT | Herb-19 |
|---|---|---|
| Vaze etal (Vaze et al., 2022a) | 35,624 | 63,901 |
| Ours | 1,192 | 1,874 |

learning to further boost the performance of our method on the task of generalized category discovery under a long-tailed distribution. Specifically, we adopt CReST (Wei et al., 2021), a popular baseline in long-tailed semi-supervised learning that adjusts the threshold for sampling different categories based on their frequency. To combine CReST with our method, we use their sampling technique to sample our $\mathcal{S}_{\text{conf}}$ set by varying the threshold for different categories. The results are presented in Tab. A3. As can be seen, introducing the long-tailed semi-supervised learning techniques into our method indeed leads to better performance.

## C  Runtime for Category Number Estimation

We compare the runtime between our method and the previous SOTA method (Vaze et al., 2022a). The result is presented in Tab. A4. Our method is more than $30\times$ faster than (Vaze et al., 2022a), while achieving more accurate category number estimation on the long-tailed datasets ImageNet-100-LT and Herb-19 (see Table 5 in the main paper).

