# OpenReview forum: "Generalized Category Discovery under the Long-Tailed Distribution"
_TMLR — Rejected by TMLR_

### Review · Reviewer_BBjY · 2024-06-10

**Summary Of Contributions:**

This paper focuses on the Generalized Category Discovery (GCD) problem, which aims to discover novel categories in unlabeled data with a long-tailed distribution. Based on an existing baseline, the authors introduce a sample selection method for training and a density-based approach for estimating the number of classes. The proposed method shows improvements in the accuracy of identifying new classes. Additionally, extensive ablation studies are presented for thorough analysis.

**Audience:**

Yes

**Broader Impact Concerns:**

None.

**Claims And Evidence:**

Yes

**Requested Changes:**

- Some parts of the method are not clear to me:
  - The proposed method is based on an existing baseline, so the authors should describe the baseline clearly. For example, I don't quite understand what the augmented view $\hat{x_i}$ and the other view $\tilde{x_i}$ are, but apparently, this is part of the proposed algorithm. Similarly, it would be better to include the objective definition of supervised contrastive learning and self-supervised contrastive learning.
  - The approach for estimating the class number is also not clear. How does Brent’s algorithm work here? What is the intuition behind applying this algorithm? Why does Brent’s algorithm address the long-tailed distribution issue? Including some equations for this part is strongly recommended.

- Regarding the experiments:
  - For Table 2, how many runs of experiments did you conduct to get the variance? How about other baselines? It seems like the improvement over ImbaGCD is not significant. Could you also report the results of a significance test?
  - For the ablation of sample selection, it is a bit surprising that only using $S_{conf}$ or $S_{dens}$ leads to a large performance drop. Do you think the number of selected examples is the reason, as they train on fewer examples? What is the ratio of the number of selected $S_{conf}$ and $S_{dens}$?
  - Also, how do you choose the hyperparameters, such as $\epsilon_k$ and $k_s$? Those hyperparameters will affect the number of selected examples. Is the proposed method sensitive to them? Providing an ablation would be helpful.

**Strengths And Weaknesses:**

- A more realistic setting compared to previous works.
- Extensive ablation studies are presented

---

### Review · Reviewer_E1ZF · 2024-06-15

**Summary Of Contributions:**

The paper addresses the problem of generalised category discovery (GCD) under long-tail distribution assumption. The aim of the task is to classify known instances and discover novel categories in a given set of unlabelled data by leveraging a set of labelled data that only spans a partial instance classes. The additional long-tail assumptions state that in the unlabelled data, the number of instances for each classes imbalanced, thus posing a siginificant challenges for triditional GCD algorithms. To address this issue, the authors proposed a sample selection and per-class instances estimation method to balance the model's training distribution. Experimental results demonstrate the effectiveness of this approach, comparing to traditional GCD methods.

**Audience:**

Yes

**Claims And Evidence:**

Yes

**Requested Changes:**

See "Weaknesses" part above

**Strengths And Weaknesses:**

Strengths:
1. Important Problem: By focusing on long-tailed distributions, the paper tackles a more realistic and practical problem, making the research highly relevant to real-world applications.
2. Clear Writing: The structure of the paper is clear, and the writing is easy to follow.

Weaknesses:
1. Novelty of the Proposed Method: At the core of this method is the sample selection method that extracts a balanced subset for model training. The use of an off-the-shelf classifier and the selection criteria using confidence or density measures can be traced in many existing pieces of literature. While the technique is quite common in long-tail recognition tasks, it is not clear how the proposed method is special to the long-tail GCD problem.
2. Thresholding: The confidence-based method relies heavily on thresholding. Determining a good threshold is very important, but this was not properly investigated in the paper.
3. Comparison to Most Related Methods: There is previous attempt to address the GCD problem under long-tail distribution using a re-weighting based method [1]. The authors should consider comparing performance with such long-tail GCD methods.

[1] Li, Z., Meinel, C., and Yang, H., 2023. Generalized Categories Discovery for Long-tailed Recognition. arXiv preprint arXiv:2401.05352.

---

### Review · Reviewer_6tXm · 2024-06-28

**Summary Of Contributions:**

The authors deal with the problem of Generalized Category Discovery (CGD) for data coming from a long-tailed distribution.

The article's contribution follows two ideas: learning the appropriate classifier and determining the precise number of classes.

The literature review is clear and thorough, and the experimental validation seems to follow the same lines as other articles in the related literature.

**Audience:**

Yes

**Claims And Evidence:**

No

**Requested Changes:**

Please see above; I have made recommendations for each point raised.

**Strengths And Weaknesses:**

I have some reservations as to the contribution, presentation and novelty of this work. I summarise my concerns as follows.

1) the article considers a series of datasets for the experimental validations, claiming they are "long-tailed". Why are these datasets long-talied? are they known to be? For the readership that might be unaware of the long-tail property of these datasets, showing (e.g., by computing the corresponding statistics) that these datasets are long-tailed might benefit the paper.

2) In general, the tables could be more informative. The captions of most of them provide little information. For instance, what are the "results" in Table 2? What's the performance indicator being shown?

3) Some formatting issues exist, particularly when table environments are used. For instance: end of page 8 / top of page 9 / top of page 10

4) Other works also deal with the problem of CGD with long-tailed distributions (in particular, the benchmarks considered by Lei et al. 2023a and Chuyu 2023.) In this regard, is the contribution of the article concentrated on the procedure defined in Secs. 3.2 and 3.3?

5) If the method is claimed to be able to deal with long-tailed data, shouldn't the results show the performance of the method wrt to a level of "long-tailedness"?

6) I failed to understand the role of the class unbalancedness in the contribution. The paper suggests that it is related to the long-trailed property of the data, but these are clearly two different concepts. Is handling unbalanced datasets a distinguishing feature of this paper? (wrt to the benchmarks)  If so, it should be stated as a contribution in a clearer way.

7) Table 1 claims to show statistics of the datasets. Perhaps more informative statistics (e.g., those showing that the datasets are in fact long-tailed) should be shown.

8) In the conclusions, the paper reads, "...our approach iteratively selects a balanced and reliable subset from the original unlabelled dataset". How can one select a "balanced" subset from an unlabelled set?

With all these points, I found it difficult to identify the paper's contribution at the bar expected at TMLR. In particular wrt to the benchmarks considered (mentioned above).

---

### Comment · Action_Editor_3pfC · 2024-06-19
**Missing review**

Dear Reviewer 6tXm,

We are still waiting for your review. Can you let me know when it would be ready? Thank you for your help!

AE

---

### Decision · Action_Editor_3pfC · 2024-08-06

**Recommendation:** Reject

**Comment:**

The submission proposed to use sample selection for generalized category discovery under the long-tail setting. The authors didn't make a good rebuttal to reply to the questions and concerns from the reviewers. Then, all the reviewers voted for rejection unfortunately. There are remaining technical concerns (though novelty is not considered as an issue). As a result, we cannot accept the submission for publication.

**Audience:**

Yes.

**Claims And Evidence:**

No, there are still technical concerns from the reviewers.